# Mobility Assessment Using Multi-Positional MRI in Children with Cranio-Vertebral Junction Anomalies

**DOI:** 10.3390/jcm12216714

**Published:** 2023-10-24

**Authors:** Flavie Grenier-Chartrand, Maxime Taverne, Syril James, Lelio Guida, Giovanna Paternoster, Klervie Loiselet, Kevin Beccaria, Volodia Dangouloff-Ros, Raphaël Levy, Timothée de Saint Denis, Thomas Blauwblomme, Roman Hossein Khonsari, Nathalie Boddaert, Sandro Benichi

**Affiliations:** 1Department of Pediatric Neurosurgery, Necker-Enfants Malades University Hospital, AP-HP, 75015 Paris, France; f.grenier-chartrand@hubruxelles.be (F.G.-C.); syril.james@aphp.fr (S.J.); lelio.guida@aphp.fr (L.G.); giovanna.paternoster@aphp.fr (G.P.); kevin.beccaria@aphp.fr (K.B.); timothee.saintdenis@aphp.fr (T.d.S.D.); thomas.blauwblomme@aphp.fr (T.B.); 2School of Medicine, Paris-Cité University, 75006 Paris, France; klervie.loiselet@aphp.fr (K.L.); roman.khonsari@aphp.fr (R.H.K.); nathalie.boddaert@aphp.fr (N.B.); 3Department of Neurosurgery, Université Libre de Bruxelles (ULB), Hôpital Universitaire de Bruxelles, CUB Hôpital Erasme, 1070 Bruxelles, Belgium; 4Craniofacial Growth and Form, Necker-Enfants Malades University Hospital, AP-HP, 75015 Paris, France; maxime.taverne@aphp.fr; 5Reference Center for Rare Diseases C-MAVEM (Chiari, Spinal Cord and Vertebral Diseases), Necker-Enfants Malades University Hospital, AP-HP, 75015 Paris, France; 6Department of Pediatric Imaging, Necker-Enfants Malades University Hospital, AP-HP, 75015 Paris, France; raphael.levy@aphp.fr; 7Department of Maxillofacial Surgery and Plastic Surgery, Necker-Enfants Malades University Hospital, AP-HP, 75015 Paris, France

**Keywords:** cervical spine, instability, multi-positional MRI, diagnosis, children

## Abstract

Objective: This study aimed to assess the relevance of using multi-positional MRI (mMRI) to identify cranio-vertebral junction (CVJ) instability in pediatric patients with CVJ anomalies while determining objective mMRI criteria to detect this condition. Material and Methods: Data from children with CVJ anomalies who underwent a mMRI between 2017 and 2021 were retrospectively reviewed. Mobility assessment using mMRI involved: (1) morphometric analysis using hierarchical clustering on principal component analysis (HCPCA) to identify clusters of patients by considering their mobility similarities, assessed through delta (Δ) values of occipito-cervical parameters measured on mMRI; and (2) morphological analysis based on dynamic geometric CVJ models and analysis of displacement vectors between flexion and extension. Receiver operating characteristics (ROC) curves were generated for occipito-cervical parameters to establish instability cut-off values. (3) Additionally, an anatomical qualitative analysis of the CVJ was performed to identify morphological criteria of instability. Results: Forty-seven patients with CVJ anomalies were included (26 females, 21 males; mean age: 10.2 years [3–18]). HCPCA identified 2 clusters: cluster №1 (stable patients, *n* = 39) and cluster №2 (unstable patients, *n* = 8). ΔpB-C2 (pB-C2 line delta) at ≥2.5 mm (AUC 0.98) and ΔBAI (Basion-axis Interval delta) ≥ 3 mm (AUC 0.97) predicted instability with 88% sensibility and 95% specificity and 88% sensitivity and 85% specificity, respectively. Geometric CVJ shape analysis differentiated patients along a continuum, from a low to a high CVJ motion that was characterized by a subluxation of C1 in the anterior direction. Qualitative analysis found correlations between instability and C2 anomalies, including fusions with C3 (body *p* = 0.032; posterior arch *p* = 0.045; inferior articular facets *p* = 0.012; lateral mass *p* = 0.029). Conclusions: We identified a cluster of pediatric patients with CVJ instability among a cohort of CVJ anomalies that were characterized by morphometric parameters with corresponding cut-off values that could serve as objective mMRI criteria. These findings warrant further validation through prospective case–control studies.

## 1. Introduction

The cranio-vertebral junction (CVJ) has a unique and distinct anatomy from the subaxial cervical spine. The CVJ is composed of the atlanto-occipital and atlanto-axial joints, strengthened by multiple ligamentous attachments and muscles to provide stability and support. In contrast to the atlanto-occipital joints, which are considered the most stable, the atlanto-axial joint is renowned for its remarkable mobility within the cranio-vertebral region and is thus more prone to developing instability [1]. 

When assessing patients with rare pediatric diseases involving the CVJ, such as mucopolysaccharidosis, skeletal dysplasia (SD), Down syndrome, syndromic craniosynostosis (SC), or the more prevalent type 1 Chiari malformation (CM-1), a recurrent question emerges: does the patient exhibit CVJ instability? This query carries significant clinical implications as it directly influences surgical planning, including the need for CVJ fixation. The identification of CVJ instability requires a comprehensive evaluation that incorporates clinical assessment, neuroimaging, and dynamic studies [2]. As outlined in two recent International Consensus Reports, there is an emerging agreement on the necessity of adopting a multidimensional approach to assess and manage CVJ anomalies [3,4]. The guidelines underscore the significance of not only radiological criteria but also clinical parameters in diagnosing and treating CVJ-related disorders.

Dynamic radiographs in neutral, flexion, and extension positions are the preferred and cost-effective imaging modality for evaluating CVJ instability. However, interpreting these images in pediatric patients can be challenging due to factors such as skeletal immaturity, bone appositions, delayed ossification, poor bone mineralization, and abnormal vertebral anatomy [5]. CT-scans provide excellent visualization of osseous structures but cause irradiation and a potential risk for developing cancer [6]. On the other hand, MRI is a valuable non-invasive tool for assessing cervical soft-tissue anomalies but does not provide dynamic information and may miss signs of spinal cord compression. 

Multi-positional MRI (mMRI), introduced in 1988 [7], allows the evaluation of structural changes during cervical motion through multi-positional imaging in neutral, flexion, and extension positions [8]. Dynamic spinal cord compression has been observed in various conditions, such as adult cervical myelopathy [9] or rheumatoid arthritis [10]. However, interpreting mMRI remains challenging as there are no established objective criteria in the literature to diagnose CVJ instability. Safety concerns regarding mMRI in children have been raised, but studies have shown that it can be safely performed even without neurosurgical supervision in very young infants with severe congenital CVJ anomalies [11,12]. 

The objective of this study was to identify CVJ instability using mMRI. To do so, we characterized CVJ instability under an unsupervised approach, established cut-off values for instability, and conducted an anatomical study of the CVJ to identify morphological criteria potentially associated with instability.

## 2. Materials and Methods

We retrospectively reviewed the medical charts and imaging data of all children who routinely underwent cervical mMRI at our institution between January 2017 and December 2021. All the included patients presented with moderate to severe CVJ anomalies and had undergone a mMRI to assess for signs of CVJ instability. Patients with poor-quality MRI images were excluded from the study. We anonymized all files, and according to local regulation, as a non-interventional retrospective study on routinely acquired data, written informed consent was waived. The following clinical parameters were recorded: age, sex, diagnosis, symptoms, and indication for surgery. Among this cohort, patients who also underwent a cervical CT-scan were identified and these data were used for the anatomical study of the CVJ.

### 2.1. mMRI Protocol

Patients underwent three sequential 2D sagittal T2 MRI acquisitions on 1.5 or 3 T devices. All mMRIs were performed without neurosurgical supervision. Acquisitions included flexion of the head, neutral position, and hyperextension of the head. For the neutral “static” sequence, the patient was placed in the supine position. For the “dynamic” sequences, a pillow (variable size according to age) was placed under the occiput to provide maximum head flexion; the maximum extension was reached by placing the pillow under the shoulders.

### 2.2. Occipito-Cervical Parameter Measurements

Twelve occipito-cervical morphometric parameters were measured on mMRI: five static occipito-cervical parameters, defined as parameters that do not change according to the position of the head and that are only measured in a neutral position, and seven dynamic occipito-cervical parameters, defined as parameters that may vary during motion, measured in the three different positions of the head. The static parameters were: (1) platybasia (angle formed by a line extending across the anterior cranial fossa to the tip of the dorsum sellae and a line along the posterior margin of the clivus) [13]; (2) Boogaard angle (angle formed by drawing a line from the plane of the clivus to the basion and a line from the basion to the opisthion) [14]; (3) tentorial angle (measured between a line connecting the nasion with the tuberculum sellae and the angle of the straight sinus) [15]; (4) C2 retroversion (measured as the angle between the base of C2 and its intersection with a line drawn from the odontoid tip) [16]; and (5) McRae’s Line (line drawn from the tip of the basion to the tip of the opisthion) [17]. The dynamic parameters were: (1) pBC2 line (pB-C2) (maximum perpendicular distance of dens to the line from the basion to the inferoposterior part of the C2 body) [18]; (2) clivo-axial angle (CXA) (angle formed at posterior border of clivus and posterior vertebral C2 line) [19]; (3) basion-axis interval (BAI) (horizontal distance between the basion and the posterior cortex of the axis) [20]; (4) basion-dens interval (BDI) (distance between the tip of the dens to the basion) [20]; (5) C1-C2 Cobb angle (measured by the angle between mid-axis of C1 and along the inferior end plate of C2 in sagittal plane) [21]; (6) Klaus height index (KI) (distance between tip of the dens and the tuberculum torcula line) [22]; and (7) C2-opisthion interval (C2OI) (distance from the tip of the dens to the opisthion) [23]. Two observers (FGC and SB) blindly performed measurements for 47 patients to assess inter-observer measurements. A single observer (FGC) assessed repeatability by performing 5 measurements in each position on 10 patients. Delta values (i.e., the absolute difference between the flexion and extension position) were calculated for each dynamic parameter (pB-C2, CXA, BAI, BDI, C1-C2 Cobb angle, KI, C2OI). Delta values (Δ) reflected the mobility of the CVJ and were used for the unsupervised clustering analysis.

### 2.3. Geometric Modeling of the CVJ

The initial phase involved the creation of a geometric model of the CVJ to assess the mobility of the bony elements based on the three-dimensional position of reference points. All mMRIs were imported into 3D Slicer v. 5.0.3. Using the segmentation tool, thirteen reference points, or landmarks, were positioned on the CVJ in a specific anatomical sequence (Figure 1) for each mMRI in flexion and extension positions. Inter-observer measurements were estimated and controlled by repeating measurements five times in one patient in the neutral, flexion, and extension positions. Intra-observer measurements were obtained by repeating measurements five times in three patients for each position. Cartesian coordinates (x, y, and z) were extracted for each reference point, and the package “morpho” in R was used to standardize the position, orientation, and size of all configurations by computing a Generalized Procrustes Analysis (GPA, using the function “ProcSym”). All subsequent analyses were based on these standardized Procrustes coordinates. The GPA aligned the landmarks corresponding to C2, meaning that C2 will be considered fixed at all times to capture the mobility of C1 and the cranial base. The displacement vectors were calculated for each landmark of C1 and cranial base in every patient to account for the bone displacements between the flexion and extension positions (Figure 1).

### 2.4. Anatomical Study of the CVJ

For the qualitative assessment of vertebral anatomy, CT-scans were segmented using Avizo v. 2019.1 (Thermo Fisher Scientific, Waltham, MA, USA) to extract 3D surfaces of C0, C1, and C2. Subsequently, the 3D surfaces were exported to Geomagic Studio 2013 for post-treatment. The 3D surfaces underwent smoothing to eliminate artificial roughness resulting from the segmentation process. Geomagic facilitated hole filling, correction of defective surface areas, and overall enhancement of the quality of the 3D images for subsequent anatomical studies. A qualitative analysis was conducted in which C0 (occipital scale, clivus, condyles), C1 (anterior arch, posterior arch, superior articular facets, inferior articular facets, transverse apophysis), and C2 (body, posterior arch, superior articular facets, inferior articular facets, odontoid, lateral mass) 3D surfaces were reviewed. The presence of normal shape/anatomy as a reference and all relevant morphological characteristics observed were documented. This qualitative analysis was performed double-blind by two authors (FGC and SB) to minimize discrepancies.

### 2.5. Statistical Analysis

As our cohort was not controlled, an unsupervised approach called hierarchical clustering based on principal component analysis (HCPCA) was used to identify patients with CVJ instability based on the delta values of the dynamic parameters measured on mMRI and the age of the patients. HCPCA is based on principal component analysis (PCA), which is a statistical technique used to analyze and reduce the dimensionality of a dataset by transforming the original correlated variables into uncorrelated variables (called principal components) that maximize the explained variance [24]. Here, HCPCA was used to identify clusters of patients based on individual resemblance. A receiver operating characteristic (ROC) curve analysis was performed to assess the accuracy of dynamic parameters for identifying unstable patients and determining cut-offs for instability. The sensitivity and specificity of these parameters were computed. A logistic regression model was used for the prediction of hypermobility in the static neutral position. A multivariate analysis of covariance was performed to assess the effect of age and clustering of patients on the displacement vectors. A Canonical Variate Analysis (CVA function in the “morpho” package) was used to identify the main mobility differences between clusters. A univariate logistic regression model was performed using the normal shape as a reference for the qualitative analysis of C0, C1, and C2. Cohen’s kappa coefficient was calculated to assess inter-observer reliability.

## 3. Results

### 3.1. Demographic and Morphometric Characteristics of the Cohort

Forty-seven children (21 males and 26 females) with a median age of 10.2 years [3–18] underwent dMRI in our center from 1 January 2017 to 31 December 2021. Diagnoses were: CM-1 (*n* = 17), SC (*n* = 10; Crouzon syndrome *n* = 6, Pfeiffer syndrome *n* = 2 and multi-suture synostosis sagittal + 2 lambdoïds *n* = 2), SD (*n* = 4), achondroplasia (*n* = 2), mucopolysaccharidosis (*n* = 3), Klippel-Feil syndrome (*n* = 3), filaminopathy type B with (*n* = 1), neurofibromatosis type 1 with C0–C1 fusion (*n* = 1), isolated foramen magnum (FM) stenosis with basilar invagination (BI) (*n* = 4), os odontoideum (*n* = 1), and C1–C2 panus (*n* = 1). Nine percent of patients were diagnosed with myelopathy, 38% had syringomyelia, and 51% presented with herniation of cerebellar tonsils. Occipito-cervical/cervical fixation for CVJ instability was performed on 5/47 patients. The means of static and dynamic morphometric occipito-cervical parameters have been compiled (Table 1). 

### 3.2. Mobility Assessment and Identification of CVJ Instability

PCA based on age and delta values of mMRI dynamic occipito-cervical parameters showed that only 5 dynamic parameters (ΔpB-C2, ΔBAI, ΔCXA, ΔBDI, and ΔKI) were significantly correlated with the mobility of the CVJ (*p*-value < 0.001). The main variability in the dataset was explained by the first two principal components (PC), which were mostly supported by ΔpB-C2 and ΔBAI (Table 2). Furthermore, the hierarchical clustering analysis clearly identified two groups in the population: Cluster №1 (*n* = 39), representing the majority of the population, was characterized by their ΔpB-C2 and ΔBAI (0.89 [0–3.5] and 1.4 [0–4.5], respectively), indicating minimal movement of the CVJ. Cluster №2 (*n* = 8) exhibited higher mean delta values of the ΔpB-C2 and ΔBAI (3.9 [2.4–6.4] and 6 [2.5–12], respectively), indicating a high level of CVJ mobility and suggesting a possible instability (Figure 2). Additionally, we did not identify any parameters predictive of CVJ mobility in the neutral static position (see Appendix A).

The analysis of the displacement vectors using a PCA and a CVA (Figure 3A) enabled the characterization of the mobility differences between the clusters (Figure 3B,C), which followed a continuum from very low mobility to instability. The stable group was characterized by a reduced flexion—extension amplitude in the movement of all bone elements. In the instable group, the extension was characterized by a great amplitude of cranial base rotation, associated with C1 in a posterior position. However, the flexion was characterized by an anterior translation of C1 compared with C2. In this dynamic position, the distance between the basion and the top anterior part of C1 remained similar, but the distances between the basion—tip of the dens, and top anterior part of C1—tip of the dens increased, suggesting a subluxation of the atlantoaxial joint.

We conducted a multivariate analysis of covariance to assess the influence of age and group categorization (cluster №1: stable patients and cluster №2: unstable patients) on the displacement vectors. Age had no significant effect (Wilk’s lambda = 0.878, F = 0.471, *p* = 0.897), but there was a significant impact of group categorization (Wilk’s lambda = 0.513, F = 3.229, *p* = 0.005) on the displacement vectors. The intersection between age and group was not significant (Wilk’s lambda = 0.665, F = 1.710, *p* = 0.118).

### 3.3. Identification of Instability Cut-Offs 

Of the five morphometric parameters correlated with CVJ mobility (pB-C2, BAI, CXA, BDI and KI), cut-offs were determined for those highly correlated with the two first principal components that discriminated the groups: ΔpB-C2 ≥ 2.5 mm and ΔBAI ≥ 3 mm were the best predictors of instability, with corresponding area under curves at 0.98 and 0.97, respectively, on ROC curves (Figure 4). Sensitivity and specificity obtained for these cut-offs were 88% and 95% for ΔpB-C2 ≥ 2.5 mm, and 88 and 85% for ΔBAI ≥ 3 mm.

### 3.4. Anatomical Qualitative Analysis of the CVJ

Among the 47 patients, 23 performed a cervical/cranial CT scan used for qualitative analysis. Cohen’s kappa coefficient was 0.61, corresponding to a good agreement between the two observers (FGC and SB). Cases in which FGC and SB initially disagreed were collectively reviewed to reach consensus. A univariate logistic regression model was performed using normal shapes as a reference for the qualitative analysis of C0, C1, and C2. C0 (occipital scale, clivus, and condyles) and C1 morphology had an effect on CVJ analysis. A significant correlation was found between the hypermobility of the CVJ and the presence of a fusion of C2 with C3 affecting different parts of the vertebra, such as the body (*p* = 0.032), posterior arch (*p* = 0.045), inferior articular facet (*p* = 0.012), and lateral mass (*p* = 0.029). Additionally, the vertical orientation of the inferior articular facets of C2 was also significantly correlated with hypermobility (*p* = 0.012).

### 3.5. Clinical Characteristics of Patients with Suspected Instability

Within cluster №2 (Table 3), CVJ anomalies were diagnosed as follows: two cases of SD, two cases of Crouzon syndrome, two cases of Klippel Feil syndrome, one case of CM-1 and one case of os odontoideum. This cluster included three female and five male patients, with a median age of 8.1 years (ranging from 5 to 12 years). Among these patients, five out of eight exhibited clinical or neurophysiological signs suggestive of myelopathy. Remarkably, even among the three patients who remained entirely asymptomatic, two had already displayed evidence of myelopathy or syringomyelia on MRI scans. Notably, six patients in this cluster were initially identified as unstable by the neurosurgical team, although only five of them underwent surgical fixation (one patient was lost to follow-up). Two patients classified within this cluster №2 were not initially regarded as unstable, including Case 3, which was a patient with Crouzon syndrome who was clinically asymptomatic but displayed mild alterations in sensitive-evoked potentials (SEP); and Case 7, a CM-1 patient who exhibited no clinical or neurophysiological signs of myelopathy. However, both cases exhibited an increase in BI during the flexion position. Additionally, Case 3 presented C2–C3 fusions, as did the other Crouzon patient, Case 4, although the latter showed mild signs of SAS with SEP alterations and subsequently underwent surgery. Cases 1 and 2 were SD patients with similar MRI characteristics, both undergoing surgical fixation for instability, despite one being asymptomatic (Case 1). Among the two Klippel Feil patients in this cluster, Case 5 remained asymptomatic but displayed myelopathy on MRI, in contrast to Case 6, who had tetrapyramidal syndrome with SEP alterations. Both exhibited similar imaging findings and presented with global C2–C3 fusions accompanied by C2 dysplasia, leading to their classification as unstable and candidates for surgical fixation. Finally, in Case 8, a patient with os odontoideum exhibited clear clinical and radiological signs of myelopathy and instability and underwent surgical fixation.

## 4. Discussion

We suggest that mMRI is a valuable tool for identifying instability in pediatric patients through the measurement of occipito-cervical parameters. We categorized patients into two clusters based on their mobility characteristics: cluster №1 included patients with minimal or negligible CVJ motion, while cluster №2 included patients with a high level of mobility, raising the hypothesis of instability. Furthermore, highly predictive cut-offs for ΔpB-C2 and ΔBAI effectively distinguished these two clusters. Although our study did not aim at identifying specific morphological criteria associated with instability, we observed a significant association between instability (cluster №2) and the presence of C2-C3 fusions. To our knowledge, this study is the first comprehensive evaluation of CVJ mobility in children using mMRI. 

We initially explored geometric modeling of the CVJ to analyze mobility and capture dynamic CVJ movements. The coordinates obtained from mMRI-generated models replicated the CVJ shapes. Based on displacement vectors, we noticed a limited overlap in the range of motion between patients with very high mobility and those with limited mobility, suggesting that mobility can be seen as a continuous spectrum of motion in patients with CVJ anomalies. High mobility in these patients was characterized by a subluxation of C1 on C2 during flexion, while the relative position between the cranial base and C1 remained similar in all three movements.

### 4.1. Current Practice and Limitations

Diagnosing instability in children with CVJ anomalies is currently challenging due to the lack of consensus definitions and assessment criteria. Instability is often defined as joint congruence, ligaments, or muscles impairment causing spinal cord compression and subsequent neurological symptoms. While hypermobility at the atlanto-occipital or atlanto-axial joints is a common indicator, the distinction between hypermobility and instability—particularly in children with chronic and progressive CVJ diseases—is an issue in clinical practice. We propose, as do other authors, that the diagnosis of instability should be based not only on imaging but on a series of clinical and radiological findings [25]. In the literature, occipito-cervical parameters are usually measured in static conditions for instability assessment. The most commonly used parameter is the atlanto-dental interval (ADI). ADI > 4 mm in flexion radiographs usually defines atlantoaxial (C1–C2) instability [5], but this value may differ regarding the series [26,27] and may have poor inter-observer reliability based on our local clinical experience. The space available for the cord (SAC) can also be used to evaluate instability. SAC has been associated with higher risks of myelopathy for SAC < 10 mm [27]. C1–C2 instability may furthermore be suggested if pB-C2 > 9 mm or CXA < 125°. CXA is used as a predictive factor for indicating CVJ fusion in CM-1 cases [28]. BAI and BDI, mostly used to diagnose atlantooccipital (C0–C1) dislocation in severe trauma, may indicate instability if >10 mm and >12 mm, respectively, on radiographs [29]. More generally, the current measurement methods for radiographic parameters may not be well-suited for pediatric cases with associated CVJ anomalies. In fact, we report that most of the mean values of dynamic parameters in static positions deviated from the normal ranges. We have introduced an innovative dynamic evaluation using the Δ values of morphometric parameters in mMRI. This method could be better suited to understand the dynamic aspects of instability by capturing and detecting hidden instabilities and providing a global assessment of stability. Furthermore, tracking changes over time offers the potential for early instability detection. 

Limited data exist regarding applicability of occipito-cervical parameters in mMRI but a recent study in adult patients has shed light on a strong correlation between occipito-cervical parameters classically used to assess instability—measured on dynamic radiographs—and multi-positional MRI, except for the ADI [30]. These results suggest the feasibility of mMRI in detecting CVJ instability, but due to limited clinical data, a comparison of parameters based on pathology type/severity was not possible [30]. In children, mMRI studies are scarce [11,12,31,32] and the demonstration of CVJ instability generally relies on indirect signs such as reduced anterior peri-medullar space [31], obstruction of CSF flow, or spinal cord compression during flexion/extension [11,12]. Our study introduces potential objective mMRI criteria. ΔpB-C2 and ΔBAI parameters were the main measurements associated with CVJ mobility and allowed to separate individuals at risk for CVJ instability into two clusters based on their motion patterns.

### 4.2. Insights of Patient Clustering and Clinical Correlation

Patients exhibiting CVJ hypermobility, indicative of potential instability, were categorized in cluster №2. It is essential to acknowledge that the diagnosis of instability should ideally encompass a comprehensive clinical assessment. Here, we examined the clinical status of these hypermobile patients to assess whether our clustering approach, primarily reliant on radiological evaluation, could reliably identify cases of instability. 

We reported 5 out of 8 patients within this cluster who displayed consistent radio-clinical indicators suggestive of instability (Table 3). Nonetheless, a subset of patients remained asymptomatic, lacking both clinical and neurophysiological signs of myelopathy, despite displaying radiological indicators of instability. For example, Case 1, diagnosed with SD, a condition characterized by bone growth disturbances leading to anomalies like odontoid hypoplasia and a high risk of instability [27,33], exhibited no initial symptoms. Initially, patients with SD may exhibit asymptomatic instability, which can progress to symptomatic cervical myelopathy and, in some cases, sudden death. C1–C2 instability is prevalent in SD, with 38% reported in one series [27] and 83% of cases presenting signs of myelopathy. Systematic flexion-extension MRI has been recommended in all patients with SD before the age of 8 to evaluate cervical stability and spinal cord compression [33].

Cases 3 and 4 had Crouzon syndrome, which is a FGFR2-related disorder associating cranio-facial malformations, hydrocephalus, and cervical spine anomalies [34]. No previous report has described an association between Crouzon syndrome and CVJ instability. A single case of atlantoaxial rotatory fixation (AARF) following a posterior cranial vault expansion has been reported [35]. Despite a history of previous posterior cranial vault expansion with C1 posterior ring resection in both our patients, there were no clinical signs of AARF. They both displayed alteration of sensitive evoked potentials, and Case 4 had mild SAS. This observation underlines the importance of meticulous assessment even in the absence of overt clinical manifestations. Also, they exhibited MRI features showing CM-1, increased BI in flexion position, and morphological anomalies on CT-scans such as C2–C3 fusions. Similar findings were observed for cases 5 and 6 with Klippel-Feil syndrome, where segmental fusions of the cervical spine are commonly found [36,37].

More generally, we report, through our qualitative analysis, a significant correlation between C2–C3 fusions and unstable patients. However, only 23 patients (49%) performed a CT scan, including 5/8 unstable patients. Despite this limitation, the implication of C2–C3 fusions seems coherent, as segmental fusion can increase the load on adjacent segments, potentially inducing adjacent segment syndrome—a phenomenon described in adult lumbar/cervical spine fusion, which may lead to spinal cord compression, cervical myelopathy, and instability [38]. Moreover, congenital fusion of C2–C3 accompanied by C1 assimilation, which can progress to BI, has already been related to a high risk of instability [39].

Among the 17 cases of CM-1, Case 7 stood out as the only patient displaying instability. Despite being asymptomatic, the patient showed an increased BI on mMRI during flexion without spinal cord compression. Recently, our team proposed a new CM-1 classification (three subgroups of CM-1 patients) based on morphological characteristics, and subgroup №2 is characterized by the presence of BI [24]. While some studies have suggested that BI is sufficient to suspect instability [40,41], the lack of tools for the evaluation of CVJ stability complicated the affirmation of this association in our previous work [24]. Based on our current results, we would recommend that CM-1 patients with BI undergo mMRI to investigate potential instability.

Lastly, Case 8 presented os odontoideum, a condition commonly associated with C1–C2 instability [42,43], and was appropriately classified in the hypermobile group. 

Finally, we observed that none of the patients who underwent surgical fixation were in cluster №1, but rather in cluster №2, the group categorized as unstable. These observations support the patient categorization we introduce, implying that the parameters and cut-off values established here could constitute relevant criteria for evaluating instability in forthcoming studies. It is crucial to emphasize that instability represents a dynamic phenomenon that can evolve over time and manifest at some point in a patient’s clinical course, transitioning from an asymptomatic state to symptomatic instability. This underscores the importance of vigilant monitoring, particularly when confronted with radiological evidence of hypermobility.

While our study primarily aimed to assess CVJ mobility rather than exploring the specifics of instability types (C0–C1, C1–C2, or both), we may assume that our patients mostly presented C1–C2 instability. It has been suggested that CVJ instability could equate to C1–C2 instability considering the rarity of C0–C1 instability, which is rather found in severe trauma or may be related to syndromic affection of multiple joints [41]. We identified that 5/8 hypermobile patients had BI, a condition often linked to chronic C1–C2 instability [41] and considered a vertical/central dislocation [1,44], leading to ventral brainstem compression. However, this relationship is not strictly one-sided, as ventral brainstem compression and instability might be present without BI [28]. Aside from the conventional parameters focused on horizontal instability (antero-posterior), there is a current lack of methods for accurately appreciating rotatory C1–C2 instability—an additional potential mechanism of instability that remained unexplored in our study due to the absence of suitable evaluation criteria.

### 4.3. Limitations and Future Prospects

Acknowledging the limitations of our monocentric retrospective approach and the lack of a healthy control group, our findings are supported by an unsupervised statistical approach that helped to control the latter bias. The sample size, representative of patients with rare CVJ malformations, was small and did not cover the full spectrum of craniovertebral anomalies encountered in pediatric populations. Additionally, the absence of a control group prevents a comprehensive distinction from being made between anomalies and variations in the normal. Furthermore, this study focuses on children, thus requiring caution when generalizing our findings to adult patients. Further prospective studies, including control groups, are needed to validate our results and pave the way for robust guidelines for diagnosing CVJ instability.

## 5. Conclusions

This study aimed to detect instability in pediatric patients with CVJ anomalies using mMRI. Our analysis categorized patients into two subgroups based on their CVJ mobility characteristics and determined morphometric parameters with corresponding cut-off values that could serve as objective mMRI criteria for diagnosing instability. While no specific morphological criteria were identified, we report that C2–C3 fusions are potentially linked to adjacent segment syndromes. These findings require further validation based on prospective case–control studies.

## Figures and Tables

**Figure 1 jcm-12-06714-f001:**
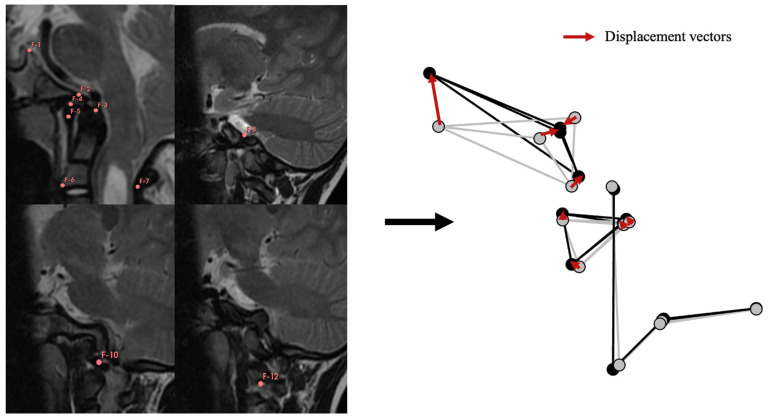
Geometric modeling of the CVJ. Reference points were positioned in the same order for each patient in each position (flexion, neutral and extension), as follows: F-1: Dorsum sellae; F-2: Basion; F-3: Tip of the dens; F-4: Antero-superior arch of C1; F-5: Antero-inferior arch of C1; F-6: Antero-inferior part of C2 body; F-7: Antero-inferior part of C2 lame; F-8: Left antero-medial part of jugular foramen; F-9 not shown but symmetric to F-8 on the right side; F-10: Left insertion site of transverse ligament; F-11: not shown but symmetric to F-10 on the right side; F-12: Left antero-supero-medial part of C2 intervertebral foramen; F-13: not shown but symmetric to F-12 on the right side. Geometric model on the right side: Flexion position shown in light grey, extension position shown in black. Using these models, “bone displacements” were calculated, represented as displacement vectors between the flexion and extension positions for each reference point (red arrows).

**Figure 2 jcm-12-06714-f002:**
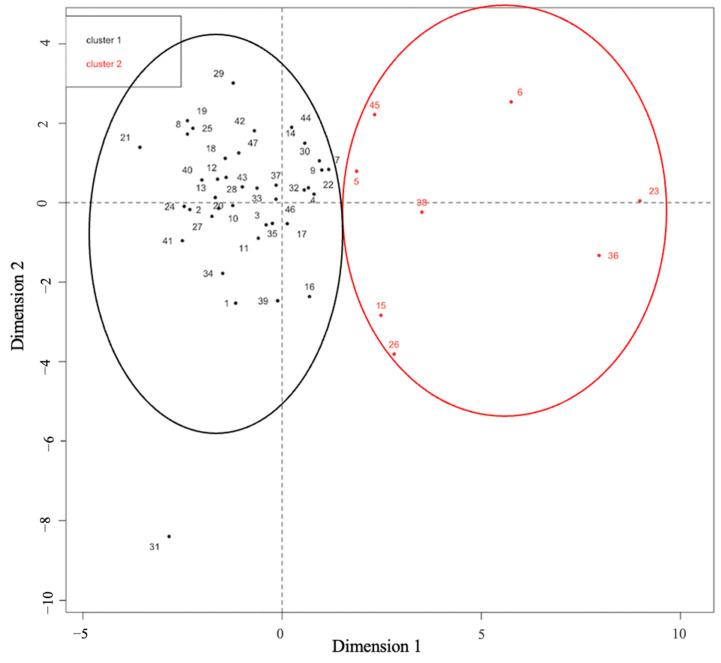
PCA with hierarchical clustering.

**Figure 3 jcm-12-06714-f003:**
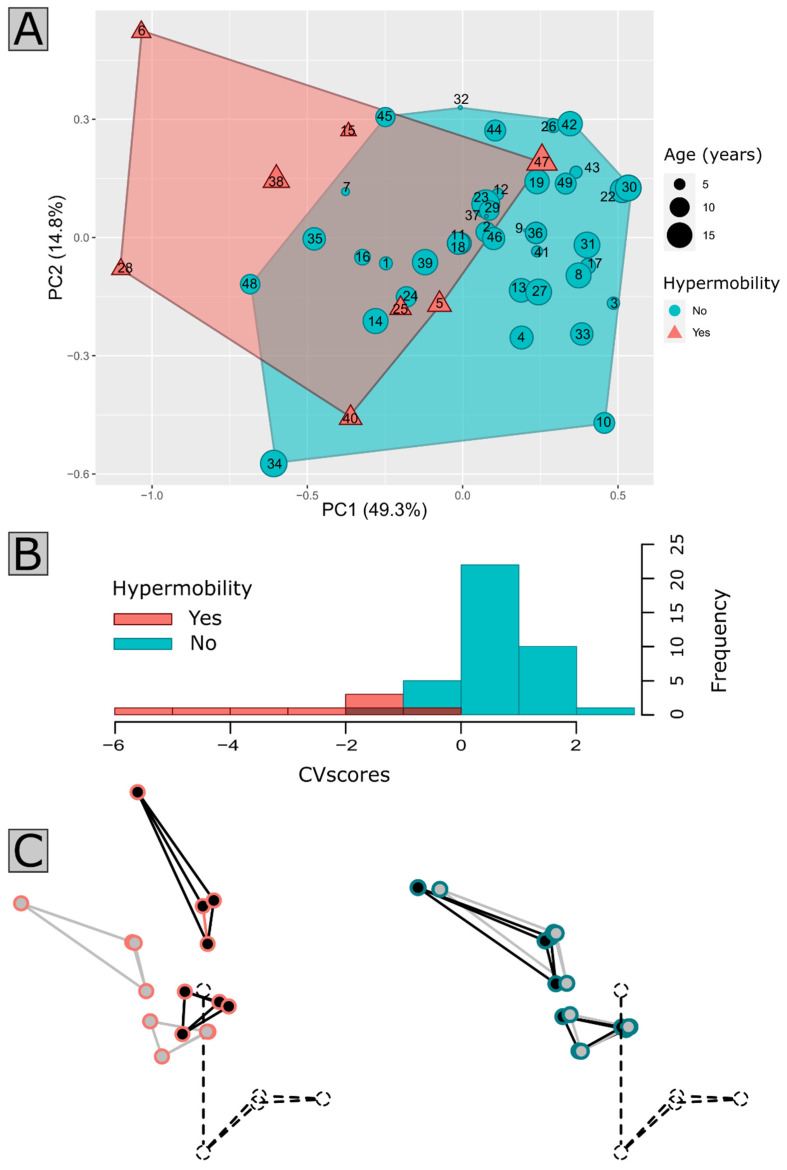
Displacement vectors analysis. Two first components of the Principal Component Analysis carried out on displacement vectors (**A**) calculated from flexion to extension configurations in all patients. Anonymized identification numbers of all the patients are indicated. Cluster 1 patients corresponding to low mobility patients, and Cluster 2 corresponding to high mobility patients are represented in cyan and magenta, respectively. The Canonical Variate Analysis (**B**) demonstrates the clear distinction between the groups along the Canonical Variable maximizing variance between groups. The theoretical shapes of the craniovertebral junction along this variable are shown in (**C**). Grey and black configurations correspond to flexion and extension, respectively. C2 is shown in dashed lines, since it was the reference for Procrustes alignment and was thus considered fixed to capture mobility in C1 and cranial base.

**Figure 4 jcm-12-06714-f004:**
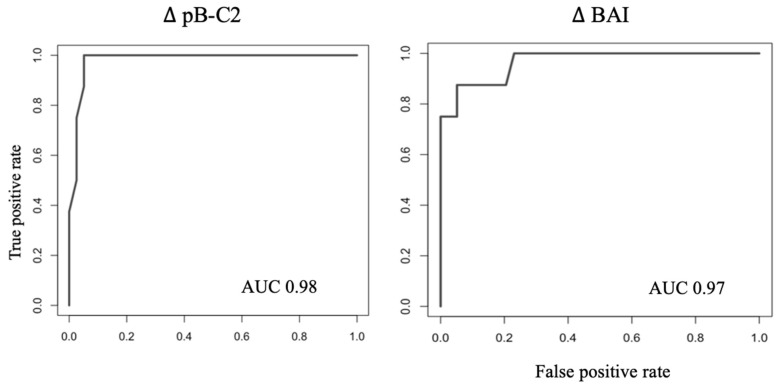
ROC curves for prediction of instability.

**Table 1 jcm-12-06714-t001:** Demographic and morphometric characteristics of the cohort.

Variables	*n* (47)
Age (years; mean [range])	10.2 [3–18]
Diagnosis	
CM-1	17
Syndromic craniosynostosis	10
Skeletal dysplasia	4
Achondroplasia	2
Mucopolysaccharidosis	3
Klippel-Feil syndrome	3
Filaminopathy type B with C0–C1 fusion	1
NF1 with C0–C1 fusion	1
FM stenosis with BI	3
Os odontoideum	1
C1–C2 panus	1
Surgery (occipito-cervical/cervical fixation)	5
MRI variables mean [range]:	
Static morphometric parameters	
Platybasia	130.7 [109–157]
Boogaard angle	137.1 [107–178]
Tentorial angle	45.2 [17–121]
C2 retroversion	74.7 [56–95]
McRae’s line	31.2 [13–41]
Dynamic morphometric parameters (neutral; flexion; extension)	
pB-C2	10.4 [6.3–19]; 10.8 [6.4–20.6]; 9.9 [4–17]
CXA	124 [82–180]; 120 [89–180]; 131 [101–180]
BAI	12.6 [5–26]; 13 [0–28]; 11 [0–22]
BDI	6.3 [0–16]; 6.2 [0–17]; 6 [0–16]
C1–C2 cobb angle	17.6 [5–50]; 14.9 [4.8–42]; 19.2 [4–40]
Klaus index	30.7 [9.3–41]; 28.7 [6.1–40]; 31,5 [8–41.7]
C2-opisthion interval	25.4 [13–37.2]; 24.7 [12.4–37]; 24.9 [10.9–34]
Qualitative MRI variables n (%)	
Myelopathy	9 (19%)
Syringomyelia	18 (38%)
Herniation of cerebellar tonsils	24 (51%)

*n*: individuals; CM-1: type 1 Chiari malformation; NF1: neurofibromatosis type 1; FM: foramen magnum; BI: basilar invagination; pB-C2: pBC2 line; CXA: clivo-axial angle; BAI: basion-axis interval; BDI: basion-dens interval.

**Table 2 jcm-12-06714-t002:** Characteristics of the Principal Component Analysis with hierarchical clustering.

			Cluster №1			Cluster №2		
	K	*p*-Value	Mean	V	*p*-Value	Mean	V	*p*-Value
*n*			39			8		
ΔpB-C2	0.6279325	3.2 × 10^−11^	0.89 [0–3.5]	/−5.374467	7.7 × 10^−8^	3.9 [2.4–6.4]	5.374467	7.7 × 10^−8^
ΔBAI	0.5627157	1.3 × 10^−9^	1.4 [0–4.5]	/−5.087723	3.6 × 10^−7^	6 [2.5–12]	5.087723	3.6 × 10^−7^
ΔCXA	0.2412186	4.6 × 10^−4^	9.3 [0.24]	/−3.331074	8.6 × 10^−4^	20.4 [4–38]	3.331074	8.6 × 10^−4^
ΔBDI	0.1803084	2.9 × 10^−3^	1.3 [0–4.5]	/−2.879963	4.0 × 10^−3^	3.1 [0.2–9.7]	2.879963	4.0 × 10^−3^
ΔKI	0.1726853	3.7 × 10^−3^	1 [0–3.7]	/−2.818426	4.8 × 10^−3^	2.8 [0.3–4.9]	2.818426	4.8 × 10^−3^

*n*: individuals; ΔpB-C2: pBC2 line delta; ΔBAI: Basion-dens interval delta; ΔCXA: clivo-axial angle delta; ΔBDI: basion-dens interval delta; ΔKI: Klaus index delta.

**Table 3 jcm-12-06714-t003:** Characteristics of the cluster №2.

Case	Sex	Age (Years)	Diagnosis	Neurological Status/Symptoms	MRI Features	CT-Scan Features + Qualitative Analysis	Fixation Surgery
1	F	9	SD	Asymptomatic	FM stenosis, increased in flexion; AMS CSF absent in flexion; odontoid hypoplasia	N/A	yes
2	M	8	SD	Pyramidal syndrome	FM stenosis, increased in flexion; AMS CSF absent in flexion; odontoid hypoplasia; myelopathy	N/A	yes
3	F	5	Crouzon	Asymptomatic; mild alteration of SEP; normal polysomnography	BI, increased in flexion; Tonsillar herniation	C2–C3 fusion (post. arch, inf. articular facet); C2 dysplasia (post. arch, inf. articular facets: lateral mass; C2 vertical inf. articular facets	no
4	M	7	Crouzon	Mild SAS and SEP alteration	BI, increased in flexion; C2–C3 and C4–C5 fusion; Tonsillar herniation	C2–C3 fusion (post. arch, inf. articular facets, lateral masses C2 dysplasia (lateral mass)	yes
5	M	6	Klippel Feil	Asymptomatic	FM stenosis and BI, increased in flexion; myelopathy	Global C2–C3 fusion C2 dysplasia (lateral mass)	Indication but lost to follow-up
6	M	10	Klippel Feil with Sprengel malformation	Tetrapyramidal syndrome; SEP alteration	BI; odontoid deformation; FM stenosis increased in flexion with anterior spinal cord compression; syringomyelia	Global C2–C3 fusion C2 dysplasia (lateral mass)	yes
7	F	8	CM-1	Asymptomatic (incidental finding); normal SEP and polysomnography	BI increased in flexion; Syringomyelia	N/A	no
8	M	12	Os odontoideum	Distal paresthesia; pyramidal syndrome	Myelopathy, C0–C1 luxation, C1–C2 cervical spinal stenosis	Odontoideum; C2 vertical inf. articular facets	yes

## Data Availability

Anonymized data shall be made available upon reasonable request to the corresponding author by a qualified investigator, after institutional review board approval.

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
