# Peer review of "Mobility Assessment Using Multi-Positional MRI in Children with Cranio-Vertebral Junction Anomalies"

_jcm, 2023, doi:10.3390/jcm12216714_

Round 1
Reviewer 1 Report
This article has many strong points in its favor: the rigorous study design, the sophisticated statistical analysis, the biomechanical modeling among them.
From this perspective, it is a true model of how an article of this kind should be written.
In simple terms, the form of the article is excellent.
On the other hand, some aspects of the substance of the article are not on par with its form.
First, the vulnerability of the joint below C2-3 congenital fusions has been a tenet of Craniocervical Junction biomechanics for several years, as frequently taught by Dr. Menezes in the past.
The 47 patients analyzed in the paper had a variety of pathologies with different biomechanical characteristics.
Although several morphometric measurements were used, their intrinsic distinction into STATIC and DYNAMIC set the stage from the start in the form of a Tautology: dynamic morphometrics will identify dynamic problems (instability/hypermobility). If so, why were the static measurements done at all?
The poor-man criterion to distinguish hypermobility from instability is the presence of movement-related signs symptoms. This article mainly focused on radiologic criteria and did not stress enough on this important clinical point.
MRI imaging was used for the measurements. The classic CCJ-related measurements dated back to the pre-CT era and relied only on bone landmarks. MRI-related morphometric measurements offer the option of HARD (bone only) and SOFT (including the ligaments) variants. The Authors did not explicitly indicate what variant they used.
Reviewer 2 Report
The authors conducted an interesting study on the relevance of using multi-positional MRI (mMRI) to identify cranio-vertebral junction (CVJ) instability in pediatric patients with CVJ anomalies, while determining objective mMRI criteria to detect this condition.
They suggested a cluster of pediatric patients with CVJ instability among a cohort of CVJ anomalies that characterized by morphometric parameters with corresponding cut-off values that could serve as objective mMRI criteria.
Overall, this manuscript is interesting and could be potential useful for objective mMRI criteria in detecting CVJ.
However, I have the following comments and suggestions, which should be solved by the authors.
1) The author should indicate important characteristics of their samples in the Abstract. Such as sample size and ages.
2) To establish a new mMRI tool in identify CVJ, samples from other centers, not only their own center, could be more reliable
3) It could be more useful to test their tool for different types of CVJ, and also, relate to the outcomes, treatment choices, would be better
4) For this wide range of age [3-18], have you considered identify ability for different ages?
5) Statistical symbols should be oblique. Such as p-values
6) What if the MRI slice did not cover the required images?
7) Also, what if the position did not match with all points?
Overall, language is readable, but there are some small format errors
Round 2
Reviewer 2 Report
Thanks for the detailed reply of the authors. They appropriately answered most of my comments and suggestions. However, I have a few more questions, which should be addressed by the authors.
1) For clinical outcomes/prediction manuscripts, an important part of it is to provide what’s the value and what this study added for clinical practice. The authors are encouraged to add more discussion on this topic. Possible references can be “Stereotactic radiofrequency thermocoagulation and resective surgery for patients with hypothalamic hamartoma. J Neurosurg. 2020 Apr 17;134(3):1019-1026”, “Characteristics, surgical outcomes, and influential factors of epilepsy in Sturge-Weber syndrome. Brain. 2022 Oct 21;145(10):3431-3443.”, “Long-term efficacy and cognitive effects of voltage-based deep brain stimulation for drug-resistant essential tremor. Clin Neurol Neurosurg. 2020 Jul;194:105940.”
Author Response
- Point-by-point response to Comments and Suggestions for Authors
Comment 1: For clinical outcomes/prediction manuscripts, an important part of it is to provide what’s the value and what this study added for clinical practice. The authors are encouraged to add more discussion on this topic. Possible references can be “Stereotactic radiofrequency thermocoagulation and resective surgery for patients with hypothalamic hamartoma. J Neurosurg. 2020 Apr 17;134(3):1019-1026”, “Characteristics, surgical outcomes, and influential factors of epilepsy in Sturge-Weber syndrome. Brain. 2022 Oct 21;145(10):3431-3443.”, “Long-term efficacy and cognitive effects of voltage-based deep brain stimulation for drug-resistant essential tremor. Clin Neurol Neurosurg. 2020 Jul;194:105940.”
Response 1: I think there is a misunderstanding, because I do not believe that these references are pertinent to the subject matter of our current study.
